# Fostering Sustainability Knowledge, Attitudes, and Behaviours through a Tutor-Supported Interdisciplinary Course in Education for Sustainable Development

**Mirjam Braßler** [1,*] and **Sandra Sprenger** [2]

1   Institute of Psychology, Universität Hamburg, 20146 Hamburg, Germany
2   Faculty of Education, Universität Hamburg, 20146 Hamburg, Germany; sandra.sprenger@uni-hamburg.de
*   Correspondence: mirjam.brassler@uni-hamburg.de

**Abstract:** Extant research into the efficacy of—especially interdisciplinary—higher education for sustainable development (HESD) is limited. A need exists to investigate students' development of sustainability knowledge, attitudes, and behaviours. Furthermore, universities have experienced difficulties implementing interdisciplinary HESD because of organisational barriers due to monodisciplinary structures, as well as educators' and students' reservations. This study introduces an interdisciplinary approach to HESD and investigates its efficacy regarding students' development of sustainability knowledge, attitudes, and behaviours at a university in Germany. The approach applies a series of lectures by different sustainability experts accompanied by several tutorials that support students' interdisciplinary learning and teamwork towards an interdisciplinary sustainability product. Tutors were trained in interdisciplinary teaching methods, as well as interdisciplinary communication and conflict management, beforehand. Before participating in the interdisciplinary course, the students had a moderate level of sustainability knowledge and behaviour, and a high level of sustainability attitudes. The results from the pre–post-test analysis indicate an increase in students' sustainability knowledge and behaviours, and no change in students' sustainability attitudes. If typical barriers to interdisciplinarity are mitigated, interdisciplinary HESD can facilitate students' development.

**Keywords:** higher education for sustainable development (HESD); project-based learning; tutor training; interdisciplinary learning; interdisciplinarity; sustainability knowledge; sustainability attitudes; sustainability behaviours

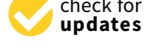



## 1. Introduction

In 2000, the United Nations (UN) Millennium Conference was held in New York City, and, as a result, the so-called Millennium Declaration with eight goals (Millennium Development Goals) was established [1]. In 2015, the 193 UN member states adopted Agenda 2030. At its core, the agenda comprises 17 global goals for sustainable development, i.e., Sustainable Development Goals (SDGs) [2]. To implement the concepts and topics in education, the UN decade for education for sustainable development (ESD) and the following Global Action Programme (GAP) [3] were launched, in which educational institutions, especially universities, aimed to implement and achieve the sustainability goals through education for sustainable development (ESD). Although all levels of the education system are basically involved in the implementation, universities play a special role in implementing the programme, as is often emphasised [4,5]. During the UN decade for ESD and GAP, various initiatives were launched, especially in higher education (HE). The formats at universities range from individual courses to whole-institution approaches, while some offer modules with a sustainability focus [6]. Several initiatives have been developed; however, so far, only limited evidence is available on the programmes' quality and efficacy in terms of knowledge, competencies, attitudes, values, and behaviour [7]. ESD

aims for students to acquire knowledge regarding sustainable development (SD) and an understanding of issues regarding their environmental, social, and economic dimensions. Furthermore, ESD strives for an individual transformation in the form of strengthening attitudes and behaviours towards SD. In order to achieve this, the literature describes (key) competencies that are to be acquired by the students. These relate to ESD in general [8], i.e., they are not limited to higher education, refer specifically to higher education [9–11], or focus on more specific areas such as teacher training [12–15]. These competencies are also covered by foundational documents of the United Nations Educational, Scientific, and Cultural Organisation (UNESCO) [7].

To achieve a holistic understanding and address complex issues, sustainability and interdisciplinarity need to be combined [16–18]. The 2030 Agenda for Sustainable Development includes global goals, such as ending poverty and hunger, protecting the planet from degradation, securing prosperity, and fostering peace, as well as global partnerships [2]. Due to their complexity, these problems cannot be solved within one academic discipline [16,19,20]. To enable students to look for relationships, interactions, and possibilities to integrate different perspectives, a need exists to implement interdisciplinary learning in higher education for sustainable development (HESD) [21–25]. Interdisciplinary learning provides an opportunity for students to gain experiences in interdisciplinary teamwork, which is necessary for both academia and employability in SD. Unfortunately, higher-education institutions (HEIs) have faced several obstacles in the implementation of interdisciplinary HESD stemming from difficulties at the organisational, educator, and student levels. The present study introduces an interdisciplinary approach to HESD that mitigates typical barriers at each level of interdisciplinarity in HE by implementing a series of lectures by different sustainability experts that present discipline-based knowledge and several tutorials that support students' interdisciplinary learning and teamwork towards an interdisciplinary product regarding sustainability. All tutors—representing a wide range of academic disciplines—were trained in interdisciplinary teaching methods, as well as interdisciplinary communication and conflict management, beforehand.

As research into interdisciplinary teaching and learning is limited [26], the present approach emphasises one particular interdisciplinary practice and its efficacy. The implementation of tutors as facilitators aims for students to experience interdisciplinary teamwork within a safe learning environment. Previous extant research into students' sustainability knowledge, attitudes, and behaviours mostly assesses the status quo within a particular academic discipline, HEI or nationwide, or how these variables interact. Most research is limited, using a cross-sectional study design with only one measurement in time. The present study's approach takes this desideratum as its starting point and shows the extent to which sustainability knowledge, attitudes, and behaviours change through a tutor-supported interdisciplinary course by applying a pre–post-test design. The present study contributes to understanding interdisciplinary learning in HESD and its efficacy regarding students' development of sustainability knowledge, attitudes, and behaviours.

## 2. Theoretical and Empirical Framework

The theoretical basis of the present study is the concept of sustainable development (SD) and, particularly, education in the context of sustainable development (ESD). Sustainable development refers to a "development that meets the needs of the present without compromising the ability of future generations to meet their own needs" [27] (p. 41).

### 2.1. Sustainability Competencies

Sustainability competencies "enable individuals to participate in socio-political processes and, hence, to move their societies towards sustainable development" [7] (p. 42), for which the expression "sustainability citizens" [28] is being used. With regard to the key competencies to be acquired, various approaches and concepts exist in the literature. In the field of education for sustainable development in general, and not just in higher education, the concept of "Gestaltungskompetenz" [8,29] is particularly widespread in

the German-speaking world. Shaping competence originally comprised a set of eight key competences [29], which were later expanded to 12 [8]. These include, e.g., foresighted thinking, as well as the interdisciplinary work relevant in this study. Barth et al. [30] took up the concept of shaping competence and examined which key competences are regarded as fundamental by students. They conducted three group discussions with a total of 13 students from the study programme "Sustainability" and from student initiatives and groups. The data show that interdisciplinary cooperation, in particular, is perceived as important. Furthermore, the study points to project-based learning as a suitable pedagogy to enable students to gain authentic skills regarding interdisciplinary teamwork.

Key competencies were also proposed for the field of higher education, which students should possess. A frequently cited framework of key competencies was developed and continued by Wiek et al. [10,11]. The authors defined five key competencies: (1) systems thinking competence, (2) future thinking (or anticipatory) competence, (3) values thinking (or normative) competence, (4) strategic (or action oriented) competence, and (5) collaboration (or interpersonal) competence. Using this framework, Brundiers et al. [31] conducted a Delphi study with 14 international experts in the field of sustainability education. The authors investigated to what extent the experts agreed with the framework of Wiek et al. [10,11] and what additions they would make. The results show that the experts agreed in principle with the Wiek framework [10,11], especially with the definitions of key competencies. The consideration of the students' later professional field is central to the consideration of competences. For example, teachers must have additional competencies, "which can be described as a teacher's capacity to help people develop sustainability competencies through a range of innovative teaching and learning practices" [7] (p.56). Therefore, specific models have been developed for the field of teacher training. These include, for example, the CSCT model (Curriculum, Sustainable development, Competences, Teacher training) [15] or the KOM-BiNE model (Kompetenzen für Bildung für Nachhaltige Entwicklung) [14]. The CSCT model is a dynamic model for ESD competences in teacher education with three levels: (1) professional dimensions, (2) the overall competencies, and (3) five competence domains [15]. The KOM-BiNE model [14] describes the following competencies, which are also related to each other: knowing, acting, feeling, valuing, communicating and reflecting, visioning, planning and organising, and networking.

Competence indicates a satisfactory state of a combination of knowledge, attitudes, and skills and the ability to apply them in a variety of situations [32]. Consequently, aiming towards students' sustainability competence development in ESD, there is a need to foster students' sustainability knowledge, sustainability attitudes, and sustainability behavioural skills, all of which are described in the next section.

### 2.2. Sustainability Knowledge, Attitudes, Perception, and Behaviours

ESD aims for students to acquire knowledge regarding SD. "With the acquisition of knowledge and information, learners come to be aware of the existence of certain realities. With critical analysis, they begin to understand the complexity of those realities." [33] (p. 4). Following the constructivist philosophy—particularly the work of Piaget, Dewey, and Vygotsky—learning is an active process in which seeking knowledge is based on personal experiences and interactions with the environment. Humans as learners perceive the world, interpret activities, and construct knowledge through questions, tests, and answers in an iterative process. Encountering a problem functions as an incentive or goal for learning and, consequently, leads to actual learning [34]. If the acquisition of new knowledge cannot be assimilated into an existing schema, a need exists for accommodation [35]. Learning across different cultures or communities allows students to co-construct knowledge aligned with social constructivism [36].

The SD concept (see above) seeks to combine issues regarding environmental, social, and economic development. Accordingly, ESD strives for knowledge development through all three pillars of sustainability and for increasing understanding of such complexities and interconnections [33]. Aiming towards development of sustainability knowledge that

addresses environmental, social, and economic issues in higher education, a need exists for an interdisciplinary approach to knowledge acquisition in ESD [21–25]. Applying the constructivist philosophy [37], interdisciplinary learning should allow students to reconstruct knowledge by reproducing knowledge from foreign disciplines, deconstruct existing knowledge by identifying one's discipline limitations, and construct knowledge by innovatively integrating ideas across disciplines [38]. Following the pragmatic–constructionist theory on interdisciplinary learning by Boix Mansilla [39], interdisciplinary understanding can increase by (1) setting an interdisciplinary purpose to guide the learning process, (2) employing disciplinary insights, (3) producing integrative understanding through the leveraging of integrations of different discipline-based knowledge, and (4) revising one's system of thought by taking a critical stance. Transforming this idea into an interdisciplinary approach in HESD, interdisciplinary knowledge development and understanding can be achieved by letting students set interdisciplinary goals regarding SD, in which they gain disciplinary insights through exposure to different discipline-based knowledge regarding SD, and then are tasked with integrating these discipline-based perspectives and taking a critical stance through reflection. All of these ideas point to the implementation of authentic interdisciplinary teamwork with actual cooperation among students stemming from different academic disciplines. This approach allows for students to model interdisciplinary teamwork within a safe learning space.

Extant research on the efficacy of interdisciplinary learning and sustainability knowledge development in ESD is limited. With regard to environmental sustainability education in Hungary, Zsóka, Szerényi, Széchy, and Kocsis [40] found a correlation between the intensity of education and environmental knowledge of students. They explained their results by pointing to the effectiveness of environmental sustainability education itself, as well as a higher intrinsic motivation of committed students who voluntarily participate in environmental education. Previous research on holistic approaches to ESD content—addressing interconnectivity among economic, social, and environmental problems—indicates a positive effect on sustainability knowledge in ESD [41–43].

In addition to aiming for students' knowledge development, ESD strives for an individual transformation, i.e., "a sustainable future start[s] with individuals and their change of behaviour, attitude, and lifestyle." [33] (p. 4). In psychology, there are several theories addressing individual behaviours and the change of these behaviours. The norm activation model (NAM) [44] and the value–belief–norm model (VBN) [45] are the two most established theories in the investigation of environmental behaviours. These theories state that behaviours are impacted by personal norms, which pressure individuals towards certain behaviours. However, the theory of planned behaviour (TPB) [46] and the reasoned action theory (TRA) as its precursor have garnered widespread support and utilisation in the context of sustainability studies. According to the theory of planned behaviour, behaviours are influenced by behavioural intentions, which, in turn, are influenced by attitudes towards the behaviours, subjective norms, and perceived behavioural control. Attitude refers to the degree to which a person offers a favourable or unfavourable evaluation of the behaviour of interest. A subjective norm refers to an individual's perception of a particular behaviour, which is influenced by judgements from significant others, such as peers, friends, educators, or family. Perceived behavioural control refers to an individual's perception of the ease or difficulty in performing the behaviour of interest. All three components are interconnected. In the educational sciences, a prominent theory addressing change of individual behaviours is social learning theory (SLT) [47]. With social learning theory, attitudes and behaviours are learned through our interactions with the social world in which we live. Incorporating interdisciplinary learning in HESD represents such a setting, enabling students to learn through social interactions with other students and educators from various academic disciplines. Participating in an interdisciplinary course on HESD could change students' behaviours towards sustainability by changing their intentions towards sustainable behaviours, thereby changing their attitudes (HESD affecting students' perceived relevance of sustainability behaviours), subjective norms

(peers, tutors or educators affecting students' perceived sustainability expectations), or perceived behavioural control (HESD expanding students' perceived opportunities to engage in sustainability behaviours).

Extant research into students' sustainability knowledge, attitudes, and behaviours mostly assessed a status quo within a particular academic discipline, HEI, or country, with some studies focussing on these variables' interactions. For example, Summers, Corney, and Childs [48] surveyed 61 students in geography and science teacher training programmes at Oxford University in the United Kingdom (UK). The authors gauged students' conceptions of sustainable development. A central result is that the respondents recognised ecological, economic, and social factors, with ecological factors mentioned most often.

The perception of students´ campus sustainability was investigated by Emanuel and Adams [49]. They compared undergraduate college students in Alabama and Hawaii, and they examined what students know about sustainability. The authors show that, in both states, about one-third did not know much about sustainability and that students were concerned about wasteful consumption of natural resources and pollution of the environment. Kagawa [50] surveyed students' understanding of and attitudes towards sustainable development. This author surveyed students in Great Britain, most of whom were studying geography. She also found that students associated the concept of sustainable development primarily with environmental aspects and less often with economic and social aspects. Biasutti and Frate [51] investigated 484 undergraduate students in Italy with high levels of sustainability attitudes, depending on academic background. With regard to the concept of sustainable living, Chaplin and Wyton [52] examined what university students living in accommodations understand about this concept and what barriers they perceive to follow sustainable living practices. In their approach, they combined qualitative and quantitative methods for an understanding of sustainability, behaviour, and the existence of a value–action gap. They were able to show that only a small percentage of respondents had an adequate understanding of sustainable living. In addition, the authors noted the perception that responsibility for obstacles was attributed to others, or that own actions made a little difference.

In the field of economics, Eagle, Low, Case, and Vandommele [53] conducted a survey in Australian undergraduate business studies and questioned business students about attitudes, beliefs, and perceptions concerning sustainability issues. The data show, for example, that responsibility for environmental problems is seen more at the government level than at the individual level. Al-Naqbi and Alshannag [54] investigated 823 university students in the United Arab Emirates and found high levels of understanding, very strong positive attitudes, and moderate behaviour towards ESD and the environment. Dominguez-Valerio, Moral-Cuadra, Medina-Viruel, and Orgaz-Agüera [55] investigated the three interaction of these three variables with students in the Dominican Republic and found no influence of knowledge towards SD on behaviours towards SD, while attitudes functioned as a mediator between knowledge and behaviour. Withley, Takahashi, Zwickle, Besley, and Lertpratchya [56] found students with biospheric and altruistic values more likely to engage in environmental behaviours than students with egocentric values. Aziz et al. [57] investigated engineering students in Malaysia, and their results indicated an influence from sustainability knowledge on sustainability attitudes.

A larger number of studies focussed on (preservice) teacher education. Cebrián and Junyent [58] explored Spanish preservice teacher-perceived ESD competencies. Among other things, participants were asked to prioritise ESD competencies in the context of a school project. The results show that student teachers prioritise the acquisition of knowledge, practical skills related to nature and natural sciences, and positive attitudes towards sustainability. Cebrián, Pascual, and Moraleda [59] investigated teacher students in Spain with results indicating a positive effect of participation in sustainability projects and students' perceived sustainability knowledge, practical skills, and actions. Keles [60] investigated 154 teacher students in Turkey and found a moderate level of attitudes to-

wards sustainability. Gündüz [61] analysed 300 university students in Libya, Nigeria, and Syria, yielding results that indicate no cultural differences in sustainability attitudes and behaviours. Borges [62] analysed 168 prospective elementary teachers in Portugal, with the results indicating high levels of sustainability knowledge and attitudes, although behaviours were less favourable than the other two dimensions. The academic discipline exerted no influence. All of these studies were limited to a cross-sectional study design with only one measurement in time.

So far, little attention has been paid to studying HESD's effects on students' knowledge, attitudes, and behaviour towards SD. A behavioural change regarding waste-prevention and sustainable travel/transportation over a 4 year period (2012–2015) was investigated by Cogut, Webster, Marans, and Callewaert [63]. The data show, for example, an increase in reports of waste prevention and sustainable travel/transportation. Tuncer [64] investigated 823 students in Turkey and found no significant mean difference in awareness of SD between students who participated in an environmental course and those who did not. Zsóka, Szerényi, Széchy, and Kocsis [40] investigated students' perception of HESD with results indicating that most students expect a positive effect of education on environmental behaviours. These positive attitudes towards HESD, as well as knowledge and action, were found to be highly interrelated with environmental education. Again, both studies also used a cross-sectional design to investigate ESD efficacy. To evaluate ESD teaching and learning, a need exists to use a pre–post-test design to address students' actual development [65]. So far, only Brody and Ryu [66] have applied a pre–post-test design to investigate an interdisciplinary course's effects from ESD and found a positive effect on students' ecological behaviour.

In summary, it can be said that there are mainly studies that describe a current situation. Evidence of findings on the effectiveness of a programme, documented, e.g., by pre–post analyses, is only available in very few cases. The present study takes this into focus and applies a pre–post-test design to investigate an interdisciplinary HESD course's efficacy on students' development regarding sustainability knowledge, attitudes, and behaviours.

*2.3. Opportunities and Challenges of Interdisciplinary Learning in ESD*

The implementation of interdisciplinary learning in ESD in higher education has become more prevalent in recent years [21–25]. Interdisciplinary learning is defined as a process by which "learners integrate information, data, techniques, tools, perspectives, concepts, and/or theories from two or more disciplines to craft products, explain phenomena, or solve problems in ways that would have been unlikely through single-disciplinary means" [39] (p. 289). Several opportunities are aligned with the implementation of interdisciplinary learning in higher education, addressing three major teaching modes at the university level: academic mode, market-driven innovation mode, and hybrid learning and responsibility mode [67,68]. Regarding academic mode, interdisciplinary learning allows students to gain a holistic view of theory and knowledge development in SD, which is a highly interdisciplinary research field [69,70] with pluralistic perspectives that stem from different discipline-based views on SD [23]. Furthermore, most students have an enviro-centric bias regarding sustainability [43], thereby neglecting the social and economical dimension. Interdisciplinary learning enables students to attain a balanced view of sustainability. Regarding the market-driven innovation mode, interdisciplinary learning enables students to work in interdisciplinary teams and gain novel ideas that are highly coveted in SD [26,71,72]. With the hybrid learning and responsibility mode, interdisciplinary learning helps students address urgent problems regarding SD that—due to their complexity—cannot be solved within one discipline [16,19]. Interdisciplinary learning enables students to look for relationships, interactions, and different perspectives across academic disciplines and to take action within their communities [24,73]. Learning across different academic disciplines enables students to gain knowledge and experience in interdisciplinary teamwork. Students' exploration and experimentation with authentic

interdisciplinarity provides a required foundation for future interdisciplinary teamwork in the field of SD.

Despite the many opportunities and advantages of interdisciplinary learning in HESD, HEIs experience several obstacles to implementation stemming from lecturers' and students' reservations, as well as organisational challenges. At the organisational level, monodisciplinary structures hinder interdisciplinary collaboration across faculties. Academic silos [23], competitiveness among disciplines [23,74,75], and structural differences [76], such as time schedules, course management systems [77], and specific discipline-mandated curricula [37], have been reported to be barriers to the incorporation of interdisciplinary teaching in HEIs. Moreover, a common obstacle is the lack of monetary incentives to cooperate [23,74], as well as difficulties in coordinating activities between faculty and the campus facilities administration [74,78]. Furthermore, successful implementation of interdisciplinary learning in ESD needs appropriate assessment possibilities, such as performance-based, formative, and multiple-source-oriented formats, e.g., reasoning exercises, practical portfolios, group-assessment tasks, and reflective journals [26]. These assessment formats usually are not covered by examination regulations across all disciplinary departments [78–80].

Both educators and students experience difficulties in interdisciplinary teaching and learning in HESD. Due to discipline-based differences in educational traditions and understandings of what comprises good teaching practices [76,81,82], lecturers experience conflicts in interdisciplinary team-teaching. Interdisciplinary conflict often originates with interdisciplinary misunderstandings [83,84]. Each discipline has its own patterns, meanings, values, knowledge traditions, codes of conduct, and ways of interacting with society [85,86]. Gupta [87] reported evaluation results from interdisciplinary learning that point to territorial issues as being the most common barriers to interdisciplinarity in the early stages. These often stem from participants' lack of understanding of other disciplines. Moreover, educators lack protocols or communication practices to facilitate dialogue on interdisciplinary ESD [19] and lack knowledge of appropriate interdisciplinary teaching methods [24]. Furthermore, choosing discipline-specific content and, thus, deciding on discipline-based trade-offs in interdisciplinary teaching in HESD have reportedly been rather difficult [75,78]. Many disciplines have their own jargon and terminology [83,84], making it cumbersome to find a shared definition of common themes or problems across disciplines [88], especially considering that each discipline has a different understanding and definition of sustainability and *SD* [75]. Thus, identifying an interdisciplinary theme in ESD proves challenging, as finding common ground is a typical barrier in interdisciplinary cooperation [18,84,89]. In addition to these different perspectives, interdisciplinary collaborations often experience conflicts due to loose agreements and diffusion of responsibility [90,91]. Educators often assume that their peers have the same work and teaching culture. However, cultures are highly distinct [22,83,84]. Furthermore, educators report having additional workloads due to coordination regarding planning, grading, and choosing appropriate topics, which are time-consuming tasks [78]. Extant research into interdisciplinary teamwork and team-teaching shows that additional work regarding interdisciplinarity is often underestimated [80,84,92,93]. Interdisciplinary team-teaching requires a change in educators' roles, which often results in educators experiencing a loss of control [26]. Furthermore, educators usually think in discipline-based stereotypes. Allocating assignments on the basis of discipline-based group affiliations and professional stereotypes is often viewed as an act of discrimination [94,95]. Moreover, profession centricity—the belief in discipline-based superiority—is also a typical barrier to interdisciplinary encounters [96].

Many barriers to interdisciplinary education regarding educators have also been reported among students [26,67,72,75,97], who also experience conflicts due to discipline-based differences. Particularly in interdisciplinary HESD, students' varying backgrounds, individual knowledge limitations in other disciplines, and a different habitus of definitions and approaches to issues in SD become apparent [75]. Students find the interdisciplinary

setting disturbing and even frightening, because it differs from the ways in which they are accustomed to learning in monodisciplinary courses [26]. Unfortunately, students experience conflicts most often in interdisciplinary project-based learning [67]—one of the most popular teaching-learning formats in HESD [25,26,72].

The present study introduces an interdisciplinary approach to HESD that mitigates typical barriers at each interdisciplinarity level in HE.

## 3. Conception of the Interdisciplinary University Course in the Context of Sustainability

To mitigate typical barriers to the implementation of interdisciplinary HESD at organisational, educator, and student levels, the present study introduces a holistic approach. At the organisational level, we implemented a series of lectures once a week and several tutorials at different times during the week to accommodate students from all disciplines. At the educator level, we refrained from incorporating interdisciplinary team-teaching in the traditional sense to avoid conflict, work overload, and feelings of loss of control. Instead, we implemented a series of lectures by different sustainability experts who presented discipline-based knowledge, as well as interdisciplinary research regarding a variety of environmental, social, and economic themes in SD. Thus, we ensured students' access to deep disciplinary insights in SD. At the student level, we incorporated several tutorials that support students' interdisciplinary learning and teamwork towards an interdisciplinary product regarding sustainability. To enhance students' interdisciplinary goal-setting, as well as production of integrative understanding across disciplines and their reflection, tutors were trained in interdisciplinary teaching methods beforehand. Furthermore, the tutors were trained in their awareness of typical challenges in interdisciplinary ESD, communication in SD, and conflict management. Therefore, tutors were able to guard against interdisciplinary conflict at the student level. The present study presents an approach that separates educators' role as transmitters of discipline-based knowledge from that of tutors as facilitators in the interdisciplinary learning process.

### 3.1. Series of Lectures by Different Sustainability Experts

The course first introduces the basics of sustainable development, with an emphasis on understanding the concept of sustainability through terms and definitions, as well as associated global challenges. The basics also include the historical development of sustainability, Agenda 2030, and the Sustainable Development Goals (SDGs). As global challenges do not stop at a subject boundary, but affect many disciplines and subjects, the concept also was oriented towards this. Therefore, the interdisciplinary concept of the lecture brings together social, natural, economic, and engineering sciences, such that an integrative perspective on global challenges becomes apparent. Thus, local, national, and global perspectives are taken into account when working on topics. In the resulting discourse, different sustainability dimensions (ecology, economy, and social issues) are brought together. Over the course of the term, lectures from different perspectives highlighted a variety of sustainability topics with reference to SDGs [2], including the following content fields:

- Biodiversity, e.g., insect mortality and sustainable forest use,
- Climate change,
- Energy, e.g., hydrogen's contribution to a sustainable energy system,
- High-performance computing,
- Thermodynamics,
- Sustainable consumption,
- Resource management,
- Business ethics and management,
- Management and sustainability,
- Lifestyle and sustainability,
- Education in the context of sustainable development (ESD).

All experts were tasked with presenting their discipline-specific SD knowledge in a manner that students from all disciplines could understand to strengthen students' disciplinary insights in SD. Furthermore, they were asked to present their interdisciplinary approaches (if possible) by combining and integrating discipline-based content. To facilitate interdisciplinary reflection, students were tasked with asking questions and implementing the presented SD knowledge through their interdisciplinary projects.

### 3.2. Tutor Training and Supervision

With the aim of preparing the tutors to lead their tutorials and support interdisciplinary student teamwork, tutor training took place at the beginning of the semester before the interdisciplinary course on sustainability began. Supervision of tutors occurred repeatedly during the semester to support them on a regular basis. All participating tutors came from different disciplines, including education, economics, sociology, biology, physics, sustainability science, geography, and political science. They were chosen on the basis of their experience in ESD and SD.

Tutor training addressed two major topics (see Table 1). The first was the enhancement of tutors' knowledge of and ability to apply interdisciplinary teaching methods. In line with the pragmatic–constructionist theory on interdisciplinary learning by Boix Mansilla [39], the tutors were trained in four areas of teaching methods related to each element of the theory: (1) methods that support identification of an interdisciplinary purpose; (2) methods that facilitate disciplinary insights in the involved disciplines; (3) methods that support the leveraging of integration of different discipline-based views on SD; (4) methods that foster a critical stance to address discipline-based limitations or to reflect interdisciplinary learning and teamwork. The second topic was the enhancement of tutors' ability to facilitate successful interdisciplinary student teamwork. Therefore, the tutors were trained in their awareness of typical challenges in interdisciplinary ESD, such as minimising stereotypes, profession centrism, and misunderstandings. Moreover, the tutors were trained on different techniques in interdisciplinary communication regarding SD and interdisciplinary conflict management.

**Table 1.** Overview of tutor training's core elements in interdisciplinary education for sustainable development (ESD).

| Topic | Areas | Methods |
|---|---|---|
| Knowledge and application of interdisciplinary teaching methods | Interdisciplinary purpose | • Identifying common themes and interests<br>• Interdisciplinary mind mapping<br>• Reading newspaper articles that address all disciplines involved<br>• Formulating goals that every discipline can identify with and addressing scientific SD-related content in each discipline |
| | Disciplinary insights | • Interdisciplinary speed dating to address discipline-based knowledge, methods, theories and assumptions relating to SD<br>• Quizzes on discipline-based technical terms relating to SD<br>• Farm alarm with different discipline-based beliefs about SD<br>• Reading scientific papers on other disciplines<br>• Explaining personal-favourite, discipline-based, scientific, SD-related content to each other in pairs |

**Table 1.** *Cont.*

| Topic | Areas | Methods |
|---|---|---|
| | Leveraging integrations | <ul><li>Multidisciplinary structuring: identifying similarities and differences in disciplinary views on SD</li><li>The Edison principle: trying to connect disciplines in any way possible (open brainstorming)</li><li>Interdisciplinary roulette, with ideas and communication on pin boards</li><li>Interdisciplinary fishbowl discussion</li><li>Interdisciplinary future workshop (three phases: critique; fantasy; implementation)</li></ul> |
| | Critical stance | <ul><li>Interdisciplinary minute paper to reflect on disciplinary boundaries and lessons learned from interdisciplinary teamwork on SD</li><li>Verbal appreciation of other disciplines and one's disciplinary limitations</li><li>Lightning round with statements regarding lessons learned from the other disciplines</li><li>Feedback after each interdisciplinary teamwork session</li></ul> |
| Facilitation of interdisciplinary student teamwork | Awareness of typical challenges in interdisciplinary ESD | <ul><li>Identifying and reflecting discipline-based stereotypes</li><li>Identifying and resolving interdisciplinary misunderstandings</li><li>Refraining from profession centrism (belief in the superiority of one's discipline)</li></ul> |
| | Interdisciplinary communication regarding SD | <ul><li>Active listening (paraphrasing, rephrasing, no interpretations)</li><li>Actively asking for explanations on foreign content</li><li>Using simple explanations of discipline-based content</li><li>Applying visualisations</li><li>Explaining discipline-based technical terms after usage</li></ul> |
| | Interdisciplinary conflict management | <ul><li>Differentiating between person and discipline</li><li>Valuing discipline-based differences as necessary pluralistic views</li><li>Discipline-neutral mediation without disciplinary favouritism</li><li>Refocussing on interdisciplinary goals and common visions regarding SD</li></ul> |

The regular supervision provided a safe space in which to reflect and share conflicts within the interdisciplinary student teams, share status and progress reports on the interdisciplinary project, answer questions regarding grading, and exchange ideas and methods regarding interdisciplinary learning.

The course took place in cooperation with the interdisciplinary Centre for a Sustainable University. The centre acts as a research network, a laboratory for innovation, and an incubator for new approaches, concepts, procedures, and methods. This centre is part of the University of Hamburg's identity as a university for a sustainable future and facilitates the university's third mission to enable students to address urgent problems regarding sustainability across disciplines [98].

### 3.3. Tutor-Supported Interdisciplinary Student Projects

All tutorials started off with the application of interdisciplinary teaching methods to enhance disciplinary insights, more precisely, getting to know each other across disciplines. Since students experience difficulties in the formation phase of interdisciplinary teamwork in project-based learning [99], this approach aims at providing a common ground across disciplines. Furthermore, the interdisciplinary teamwork was guided by following steps: (1) formulation of an interdisciplinary project, (2) clarification of terms and concepts throughout the disciplines, (3) brainstorming about monodisciplinary theories, models, and methods related to the problem, (4) classification and structuring of brainstorming by identifying possible connections and discrepancies throughout the disciplines, (5) formulation of interdisciplinary goals, (6) self-study by reading papers across disciplines, (7) identification of interdisciplinary strategies and solutions, and (8) application of solution. Furthermore, (local) possibilities for action were identified and reflected on as a function of the presentations. The tutors functioned as a safety net if interdisciplinary conflict in the student teams arose. Since the tutors stemmed from various academic disciplines, they were able to model interdisciplinary communication, integration, and conflict resolution. In a final event (exhibition) at the end of the course, the results were made available to the public. Some products were related to individual lectures, while others were related to several lectures or to a broader issue in the sustainability context. The range of products presented was very diverse, with the various projects listed below.

- Caricature and short story on the topic "Virtual Water";
- Education for Sustainable Development: Which factors influence the success of ESD?
- Film production: Interviews with experts on sustainability;
- Game: "Trivial Sustainability";
- Guide to a sustainability working group (for schools);
- Insect hotel with flowering meadow;
- Model of a sustainably designed schoolyard;
- Poster: Population development;
- Poster: Sustainable Development Goals, e.g., fewer inequities;
- Project Week for Sustainable Food: What ends up on my plate?
- Sustainable economy: Development of a sustainable school currency;
- Sustainability: What are we already doing, and what hinders us from doing even more?
- Sustainable dissemination of information and discourse;
- Sustainable Production of Food in Agriculture: Permaculture;
- Sustainable school garden;
- Sustainable urban development: To what extent do urban development projects in Hamburg meet current sustainability goals?
- School subject: "Future";
- Sustainability website.

Most students presented their ideas and concepts on posters, while others worked with handicrafts, e.g., building an insect hotel. Some created a product in the form of a video or website. During the final presentation at the end of the term, participants and guests awarded prizes to the best products. The three winners are described in more detail below.

Student project: "Insect hotel with flowering meadow"

This product was based on a lecture by a zoologist who spoke about facts and causes of the great insect mortality. These students investigated the question of how the topic of insect mortality can be addressed in school and what activities can be implemented in the school setting. In addition, the student group built a large, wooden insect hotel, which they displayed at the exhibition, and presented a concept for integration into everyday school life. This concept provided for the planning and construction of the schoolyard together with the students, which made a multitude of experiences and observations possible, as

well as implementation of primary and secondary education teaching projects, including a school bee colony and a beekeeping course.

Student project: "School subject: 'Future'"

In contrast to the existing subjects, the aim of the new innovative school subject "Future" is to offer a broader range of teaching content at various levels. The subject sees its task in the dialogue between the individual and the social reality with regard to overarching sustainability and globalisation. The school subject "Future" is also based heavily on SDGs to contribute to education for the future. The idea is to develop a curriculum with an emphasis on acquiring competencies, with the overall goal of shaping the future. Ideally, the lessons will be project-oriented and taught in an interdisciplinary way, based on elements of self-directed learning supported through excursions, simulations, or offers, such as future workshops. Schools should create appropriate teaching and learning frameworks to make such cross-curricular learning possible. Another part of the concept is cooperation with partners from the region.

Student project: "Model of a sustainably designed schoolyard"

This student project's guiding question was as follows: What possibilities does school-yard planning offer for implementing the objectives of the concept of "education for sustainable development"? The group developed proposals in three areas: possibilities for waste separation and recycling, integration of a school garden, and possibilities in the area of physical activity and discovery landscapes, including (nature-oriented) playgrounds.

### 3.4. Degree Programmes

The students came from very diverse academic backgrounds (listed below), ranging from the natural and social sciences to interdisciplinary study programmes.

Business Administration (B.A.),
Educational Sciences (M.A.),
Geography (B.Sc.),
Human-Computer Interaction (B.Sc.),
Human Movement Science (B.A.),
Informatics (B.Sc.),
Linguistics (B.A.),
Multilingual Educational Linguistics (M.A.),
Political Science (B.A.),
Psychology (B.Sc.),
Social and Cultural Anthropology (B.A.),
Sociology (B.A.),
Teacher training in primary and lower secondary education (M.Ed.),
Teacher training in secondary education (M.Ed.),
General lectures,
Part-time courses.

This structure allowed for interdisciplinary consideration of the topics, especially in the tutorials. Within their degree programme, students could choose the course; for some, it was a voluntary course, while, for most others, the course was chosen from a compulsory group.

## 4. Materials and Methods

### 4.1. Participants

A total of 68 German-speaking students (female = 69.6%, male = 29.0%, diverse = 1.4%; age mean (M) = 25.07, SD = 6.86) participated in the interdisciplinary course. Both the attendance of the interdisciplinary lectures and the tutorial were mandatory. The average number of higher-education semesters among the participants was M = 3.48 semesters (SD = 3.26). Furthermore, 13% of the students studied natural sciences, 79.75% studied

educational and social sciences, and 5.8% studied the humanities. Of the 68 students, 45 were measured twice (at baseline and post course).

*4.2. Materials*

A paper-and-pencil questionnaire measured students' sustainability knowledge, sustainability behaviours and attitudes, and demographic data on age, gender, and higher education semesters.

### 4.2.1. Sustainability Knowledge

To assess sustainability knowledge, we developed a five-item self-report instrument with a five-point Likert scale answer format ranging from 1 ("strongly disagree") to 5 ("strongly agree"). The items used were the following: "I am familiar with social sustainability issues", "I am familiar with ecological sustainability issues", "I am familiar with economic sustainability issues", "I know how my discipline can contribute to sustainability", and "I know theories, models and methods in my discipline that contribute to sustainability". Reliability was viewed as acceptable for both the baseline and post-course measurements, with an internal consistency of $\alpha_1 = 0.77$ and $\alpha_2 = 0.70$, respectively.

### 4.2.2. Sustainability Behaviours and Attitudes

Both sustainability behaviours and attitudes were measured using a standardised test developed by the International Institute for Sustainable Development [100]. All items on the 15-item subscale "behaviours concerning sustainable development" (example item: "I walk or bike to places instead of going by car") and the 15-item subscale "attitudes towards sustainable development" (example item: "sustainable development will not be possible until wealthier nations stop exploiting the labour and natural resources of poorer countries") were answered on a five-point Likert scale ranging from 1 ("strongly disagree") to 5 ("strongly agree"). Following the translation and adaptation guidelines by Hambeleton and de Jong [101], all items were translated into German and then back into English, so that three native speakers could compare the original and reverse translations on literal and contextual equivalence with satisfying results (all over 80%). Both scales were viewed as having good face validity, although checking further validity dimensions was severely limited because of discord in defining the construct [100]. The reliability of the subscale "behaviours concerning sustainable development" was viewed as acceptable for both the baseline and post-course measurements, with an internal consistency of $\alpha_1 = 0.67$ and $\alpha_2 = 0.69$, respectively. The reliability of the subscale "attitudes towards sustainable development" was also viewed as acceptable, with an internal consistency of $\alpha_1 = \alpha_2 = 0.74$.

## 5. Results

A missing-values analysis indicated that Little's [102] missing completely at random (MCAR) test result was insignificant ($\chi^2 = 21.1$, df = 17, $p = 0.221$). When significant, this test suggests that the hypothesis that the data are MCAR can be rejected. Therefore, no evidence suggested that the data were not MCAR. As such, pairwise deletion was used in the statistical analyses.

Table 2 shows the means, standard deviations, and Bonferroni correction *t*-test results for time 1 (before the course) and time 2 (after the course) on students' sustainability knowledge, behaviours, and attitudes.

**Table 2.** Results from the Bonferroni correction *t*-tests.

| | *N* | $M_1$ | $SD_1$ | $M_2$ | $SD_2$ | 95% CI LL | 95% CI UL | *t* | *df* | *d* |
|---|---|---|---|---|---|---|---|---|---|---|
| Sustainability knowledge | 45 | 3.06 | 0.72 | 3.66 | 0.64 | −0.84 | −0.36 | −5.02 *** | 44 | 0.748 |
| Sustainability attitudes | 46 | 4.59 | 0.30 | 4.61 | 0.36 | −0.09 | 0.06 | −0.40 | 45 | 0.080 |
| Sustainability behaviours | 46 | 3.75 | 0.50 | 4.01 | 0.47 | −0.38 | −0.15 | −4.52 *** | 45 | 0.666 |

Notes. *** $p < 0.001$. *N* represents the sample size. $M_1$ and $M_2$ represent the means, while $SD_1$ and $SD_2$ represent the standard deviations for time 1 (before the course) and time 2 (after the course). The Bonferroni correction *t*-test results *t* are presented with effect estimates with lower limits (*LL*) and upper limits (*UL*) of the 95% confidence interval (CI), degrees of freedom (*df*), and Cohen's effect size (*d*).

The descriptive data indicate a moderate level of sustainability knowledge and behaviour, and a high level of sustainability attitudes before participation in the interdisciplinary course. The *t*-test on sustainability knowledge shows a significant increase from 3.06 (pre-test) to 3.66 (post-test), with a large effect size at the end of the term, compared with the beginning of the term. The *t*-test on sustainability behaviours shows a significant increase from 3.75 to 4.01, with a large effect size at the end of the term compared with the beginning of the term. No significant increase was found in sustainability attitudes.

## 6. Discussion

Universities play an important role in the implementation of education for sustainable development [4,5]. A large number of initiatives have been developed; however, so far, only limited evidence is available on the quality of programmes and their efficacy in terms of knowledge, competencies, attitudes, values, and behaviour [7]. Moreover, most researchers used a cross-sectional study design, thereby limiting research to only one measurement in time.

Furthermore, a need exists for an interdisciplinary approach to address the complexity of environmental, social, and economic perspectives in HESD [21–25]. Unfortunately, HEIs experience several obstacles on the road to implementation of interdisciplinary HESD stemming from difficulties at the organisational, educator, and student levels. As shown in the Section 2, present studies describe particularly a current situation. Evidence of findings on the effectiveness of a programme, documented, e.g., by pre–post analyses, is only available in very few cases. The added value of the study is the evaluation of the effect of a measure through a pre–post comparison, which is rarely found in the studies mentioned above (Section 2). The present study applies pre–post-test design to investigate the extent to which sustainability knowledge, attitudes, and behaviours change through a tutor-supported interdisciplinary course, which was specially designed to mitigate typical barriers to interdisciplinarity in HE at the organisational, educator, and student levels. A series of lectures by different sustainability experts aimed to present discipline-based knowledge, while several tutorials focused on supporting students' interdisciplinary learning and teamwork to create interdisciplinary products that promoted sustainability. The role of the tutors was to support and facilitate interdisciplinary student teamwork. By providing an example of interdisciplinary communication and integration, students could follow their guidance. The tutors provided a safe space for students to experience interdisciplinary teamwork and its challenges. This way, the implementation of interdisciplinary project-based learning provided an opportunity to model interdisciplinary teamwork.

This study's results provide an overview of the status quo on students' sustainability knowledge, attitudes, and behaviours before course participation, as well as any changes regarding these three variables over time, while deriving implications for the design of interdisciplinary teaching and learning in HESD. Before participating in the interdisciplinary course, the students had a moderate level of sustainability knowledge and behaviour, and a high level of sustainability attitudes. Previous studies on knowledge regarding sustainability show a different situation. While some studies (e.g., Emanuel and Adams [49]; Chaplin and Wyton [52]) indicate that, for a part of the students, little or insufficient knowl-

edge about sustainability is present, other studies point to high levels of knowledge in this field [54,62]. The results of these studies indicating moderate levels of sustainability knowledge contradict some previous findings indicating high levels of knowledge and understanding among students in Portugal [62] and in the United Arab Emirates [54]. The findings of the present study tend to be more in the direction of Emanuel and Adams [49] and Chaplin and Wyton [52] or rather moderate knowledge. However, a direct comparison of the studies can only be made to a limited extent, since the design and the measuring instruments used are very different.

The results indicating strong positive attitudes towards SD support recent findings by Biassuti and Frate [51], Borges [62], and Al-Naqbi and Alshannag [54]. High levels of positive attitudes towards SD are in line with recent developments that concern increasing waves of youths protesting climate change and who are aligned with awareness-raising campaigns [103]. As the course is an optional one among a larger number of others that students could choose from, a possible explanation for the high values for attitudes, therefore, would be an already-existing interest in the topic. The results that indicate moderate sustainability behaviours in students also support recent findings [54,62] that suggest a high level of sustainability attitude, accompanied by moderate levels of sustainability behaviours, further establishing the well-known attitude–action gap in the student community.

Data from the pre–post-test analysis indicate an increase in students' sustainability knowledge and behaviours, and no change in students' sustainability attitudes. The increase in students' sustainability knowledge and behaviours indicates a positive effect from the interdisciplinary course on students' development. The increase in sustainability knowledge in all three dimensions (environmental, social, and economic) of development corresponds with constructivist philosophy. Furthermore, the results support the proposed advantage of an interdisciplinary approach to HESD. The implementation of core elements (interdisciplinary purpose, disciplinary insights, leveraging integrations, critical stance) of the pragmatic–constructionist theory on interdisciplinary learning [39] allowed students to address and integrate social, environmental, and economic perspectives on SD. The results are, thus, in line with the study by Barth et al. [30], which demonstrated the special importance of interdisciplinarity and the application of project-based learning. They examined which key competencies could be considered fundamental; their data show that interdisciplinary cooperation, in particular, appears to be central. The benefits from this holistic approach support previous research highlighting a positive effect on sustainability knowledge in ESD [41,42].

The results from the pre–post-test analysis indicating an increase in sustainability behaviour throughout the semester build on previous research [66]. While Brody and Ruys' [66] findings suggest further development in ecological behaviours through an interdisciplinary ESD course, the present study indicates an increase in sustainability behaviour in all three dimensions, including social and economic. Regarding the theory of planned behaviour (TPB) [46], changes in sustainability behaviour indicate changes in students' intentions towards sustainable behaviours. Following the theory, this could be the result of an increase in students' attitudes, subjective norms, or perceived behavioural control. Since the pre–post-test analysis indicates no change in sustainability attitude, one can assume—in turn—a change in students' subjective norms or perceived behavioural control. More precisely, participation in the interdisciplinary course might have affected the students in two possible ways. First, students' perceived sustainability expectations might elicit changes through interactions with peers, tutors, or educators. Second, students' perceived opportunities to engage in sustainability behaviours might have changed through engagement in the interdisciplinary project. Despite this possible channel, these results strongly promote social learning theory and the importance of learning through interactions with the social world in which the students live. Lastly, the result indicating no change in students' sustainability attitude is worth discussing. Even though one might assume that an attitude change based on TPB with HESD potentially affects students' perceived relevance of sustainability behaviours, the students assessed in the present study already

had high levels of sustainability attitude before course participation, leaving little scope for further development.

This study's results point to the advantages of the implementation of interdisciplinary learning in HESD. The increase in sustainability knowledge and behaviours indicates successful designing of an interdisciplinary teaching–learning environment. Thus, the results indicate that the separation of roles—with educators as transmitters of discipline-based knowledge and tutors as facilitators of the interdisciplinary learning process—worked in favour of students' development. The tutors were able to support students' processes of interdisciplinary goal setting, production of integrative understanding across disciplines, and their reflection towards their sustainability products. This way, many typical barriers to interdisciplinarity in HESD could be mitigated. In this design, severe conflicts due to discipline-based differences in interdisciplinary student and educator teams could be avoided. However, many typical difficulties regarding interdisciplinarity remain a challenge. Particularly at the organisational level, we experienced barriers regarding different examination regulations across faculties because there actually would have been different credit points, depending on their integration into different degree programmes or courses from different disciplines. Therefore, at the beginning of the course, the credit points had to be standardised to avoid inequities and injustices within the course. Furthermore, we experienced difficulties in incorporating the interdisciplinary course within the discipline-based curricula. For example, preservice teachers could choose the interdisciplinary course within their discipline-based curriculum, while the course was creditable only in the general studies segment for students from other academic disciplines. Furthermore, the financing of interdisciplinary teaching remains a central challenge. All experts, most of whom were professors, shared their knowledge without any payment in the form of a teaching allowance or honorarium. They all did their 90 min presentations in the service of SD or as a personal favour. Some tutors were paid by faculties, while others were paid by an institution outside the faculties or the interdisciplinary Centre for a Sustainable University. To further facilitate the implementation of interdisciplinary learning in HE, a need exists for HEI to follow a whole-university approach towards an interdisciplinary university by design to overcome institutional barriers to interdisciplinary learning in ESD [74]. At the organisational level, HEIs could implement interdisciplinary learning spaces, in-house training for educators, university-wide financial support of interdisciplinary HESD, and the use of economics of scale across university departments in regard to resources, such as space and administrative support [77]. All faculties involved in interdisciplinary HESD could easily share the expense of paying for the tutors.

This study contains several limitations. Due to shortening the questionnaire, we developed an instrument for measuring sustainability knowledge; thus, it is not standardised. Furthermore, there is no academic agreement on the use of an established instrument that measures SD attitudes, as the concept derives from different theoretical backgrounds that underlie the various approaches. Even though Michalos et al.'s [100] instrument is based on the three dimensions—environment, economy, and society—it might lack an emphasis on education. The approach by Biasutti and Frate [51] indicates a structure of four areas, addressing the environment, economy, society, and education regarding attitudes towards SD. These results indicate that attitudes towards SD might be more complex than the present study's approach suggests. Moreover, their approach indicates differences between psychology and agriculture students. Consequently, future research should also further investigate differences among students stemming from their varied academic backgrounds. This is especially important for interdisciplinary course evaluation, as participation could affect students differently. Furthermore, recent research into students' attitudes and behaviours showed the importance of personality traits [104] which, in turn, should also be addressed as potential mediators in pre–post-test measurements in educational settings.

High levels of positive attitudes towards SD before participating in the interdisciplinary course on sustainability might indicate a selection bias among students who chose this particular course on the basis of their positive attitudes and accompanying interests in



SD. Consequently, future research should investigate a variety of students representing different levels of sustainability attitudes. Furthermore, different types of attitudes such as attitudes towards sustainability issues [105], as well as attitude shaping [40], and the effect of HESD on those might be a fruitful approach. Furthermore, the study was also restricted due to the limited number of students participating in one interdisciplinary course. Moreover, the participants attended the same university in Germany, limiting the study's generalisability.

The theory of planned behaviour is limited to explaining intentional behaviours and, consequently, lacks explanations for unintentional behaviours that could play a role in sustainability behaviours. Moreover, future research should investigate all variables of the TPB in HESD settings to gain further insights into interconnections and development in educational sustainability settings. Furthermore, a combination of the variables of the TPB with the variables of the value–belief–norm model might allow for a holistic investigation of HESD effects on students' behavioural change.

If we look not only at the quantitative data from the pre–post-test analysis, but also at the products developed by the students, we see that many of the key competences (see Section 2) of education for sustainable development have been taken up and applied. This can be seen, for example, in the product "School subject 'Future'", which reflects elements of Gestaltungskompetenz [8,29] such as foresighted thinking. In order to further underline this, further qualitative analysis would be conceivable and necessary.

Following Redman, Wiek, and Barth's [65] advice, future research regarding ESD's efficacy should combine a pre–post-test design with other assessments. Particularly in the context of interdisciplinary HESD, future research should further investigate tutors' efficacy by measuring their application of interdisciplinary teaching methods, communication, and conflict management, as well as their influence on student learning.

**Author Contributions:** Conceptualization, M.B. and S.S.; methodology, M.B.; software, SPSS 26., M.B.; data curation, M.B.; writing—original draft preparation, M.B. and S.S.; writing—review and editing, M.B. and S.S.; project administration, S.S. and M.B. Both authors have read and agreed to the published version of the manuscript.

**Funding:** This research received no external funding.

**Institutional Review Board Statement:** Ethical review and approval were waived for this study by the department of work and organizational psychology at the University of Hamburg.

**Informed Consent Statement:** Informed consent was obtained from all subjects involved in the study.

**Data Availability Statement:** The data set is not available.

**Acknowledgments:** The authors thank the interdisciplinary Centre for a Sustainable University, especially Hilmar Westholm, for its support.

**Conflicts of Interest:** The authors declare no conflict of interest.

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
