# Peer review of "Fostering Sustainability Knowledge, Attitudes, and Behaviours through a Tutor-Supported Interdisciplinary Course in Education for Sustainable Development"

_sustainability, doi:10.3390/su13063494_

Round 1

Reviewer 1 Report

Dear authors, your paper is sound and very valuable in its contents and definitions. I have only two 'small'comments to address:

1) I suggest removing the parenthesis in the title.

2) Please define in the legend of Table 2, all the parameters.

Author Response

thank you so much for your valuable time and useful contribution. We highly appreciate the inputs that definitely helped to improve our article.  We removed the parenthesis in the title. We defined all parameters of table 2 in the legend (line 606-608). Thank you much for pointing us to this direction.

Reviewer 2 Report

The general context is well presented in the Introduction section.

I appreciate the intensive/highlighted focus on student presenting educational models/teaching modes at the university level (ll. 263-274).

Subject/Discipline integration is an intensively discussed topic, STE(A), CLIL, ITI and the research shows both, positive and negative aspects/ strengths and weaknesses. The present article provides and discusses the opportunities integration of ESD into different disciplines can bring.

I would suggest changing the tone of the sentence “Every educator assumes that the other has the same work and teaching culture.“ (l.306)

The reader can find a brief information on teachers, similarly about the students, their study programmes. If it is possible and the authors have data, it might be useful to know the age of students as well as the proportion of students according to their main study programmes. There are 16 study programmes mentioned including e.g. Geography, Social and Cultural Anthropology, linguistically oriented programmes, etc.  Similarly, the information on the number of students who sign the course voluntarily and the numbers of students attending the compulsory course might be interesting information.

I would also suggest the authors to make a separate paragraph Limitations of the study and concentrate them at one place (one paragraph) - the limitations are described as a part of discussion.

Minor language mistakes that do not affect understanding of the text can be found in an article, e.g

108-109 Specific models have therefore developed for the field of teacher training.

Line 154 voulatrily

Line 243 on emvironmental

Line 244 attitides

Line 269 on interdisciplinary teams (in?)

Line 386 methods that fostered (you may use present tense as in the 3 previous items you use the present tense)

Line 649 Due to shorten the questionnaire

Author Response

This manuscript is a resubmission of an earlier submission. The following is a list of the peer review reports and author responses from that submission.

Round 1

Reviewer 1 Report

Presented article describes research about sustainability interdisciplinary course in German. It is very important topic. Article entiteled "Fostering sustainability knowledge, attitudes and behaviour through a tutor-supported interdisciplinary course in (education for) sustainable development" well written, clearly structured. There are few issues that authors, in my opinion, have to consider and justify, as well as some that they might take under consideration.

  1. I think there is a mistake on page 1, line 39 and line 43: correct is Global Action Programme (GAP).
  2. I would reccomend that you add a reference to the definition od SD on page 2, line 92. (Brundtland commission)
  3. Consider changing word "increasing" into "changing" on page 3, line 129.
  4. On page 5, line 215, consider changing word "each discipline" into "many disciplines". In my oppinion it is not necessary that all have digfferent view on the issue.
  5. On page 10, Participants, lines 397-401, please explain if all participating students that were measured twice attended all the lectures of the course. Was the course attendance mandatory etc.
  6. Please explain effect size for attitudes in Table 2 on page 11, why you did not provide number as according to APA standards?
  7. Please consider adding references for the last statement in line 497, page 12.
  8. Overall, very good job.

Author Response

Dear Reviewer 1,

thank you so much for your valuable time and useful contribution. We highly appreciate the inputs that definitely helped to improve our article.

We provide a point-by-point response to your comments in the following.

  1. I think there is a mistake on page 1, line 39 and line 43: correct is Global Action Programme (GAP).

Yes, indeed, it was an unintentional mistake, we changed it to GAP. Thank you very much for reading it carefully.

  1. I would reccomend that you add a reference to the definition od SD on page 2, line 92. (Brundtland commission)

We have included the definition from the Brundtland commission at the beginning of theory chapter 2 (line 91 ff.), to which we refer again later in the text.

  1. Consider changing word "increasing" into "changing" on page 3, line 129.

Thank you. We changed the wording according to your advice.

  1. On page 5, line 215, consider changing word "each discipline" into "many disciplines". In my oppinion it is not necessary that all have digfferent view on the issue.

Thank you. We changed the wording according to your advice.

  1. On page 10, Participants, lines 397-401, please explain if all participating students that were measured twice attended all the lectures of the course. Was the course attendance mandatory etc.

Thank you for pointing us to missing the information regarding attendance. We added “Both the attendance of the interdisciplinary lectures and the tutorial were mandatory.” in the Participants section in “Materials and Methods”.

  1. Please explain effect size for attitudes in Table 2 on page 11, why you did not provide number as according to APA standards?

Thank you for pointing this out. We left out the effect size because we did not find a significant effect in attitudes. We added the effect size now. Thank you!

  1. Please consider adding references for the last statement in line 497, page 12.

Thank you. We added the reference in the statement.

  1. Overall, very good job.

Thank you very much!

Reviewer 2 Report

This a well constructed manuscript dealing with research on sustainability knowledge, behaviour and attitudes. The manuscript is sound and well structured and athough the results are short, the findings are well discussed. I have minors remarks to be addressed to the authors:

1) In the Introduction section I am missing a more concise and defined objective. Now the specification of the objective is to generalist and vague.

2) Throughout the manuscript you state that the dimensions of analysis are sustainability knowledge, behaviour and attitudes. To me the way the dimensions are written is quite confuse.  To me is quite difficult to undersand, for example, what is sustainability attitude. I recommend the authors to think on redifining the terms. I would propose, instead, knowledge on sustainability, and behaviour and attitudes towards sustainability.

Author Response

Dear Reviewer 2,

thank you so much for your valuable time and useful contribution. We highly appreciate the inputs that definitely helped to improve our article.

We provide a point-by-point response to your comments in the following.

This a well constructed manuscript dealing with research on sustainability knowledge, behaviour and attitudes. The manuscript is sound and well structured and athough the results are short, the findings are well discussed. I have minors remarks to be addressed to the authors:

1) In the Introduction section I am missing a more concise and defined objective. Now the specification of the objective is to generalist and vague.

Thank you. We followed your advice and hope that these changes meet your idea. First, we shortened the general introduction to the history of the SDGs and their status quo to promptly emphasize our objective: interdisciplinary learning in HESD and its efficacy in enhancing students’ knowledge regarding sustainability, and students’ behaviour and attitudes towards sustainability. Second, we expanded our introduction and explanation of our objective. Third, we rearranged the explanation of the research approach to strengthen our argumentation.

2) Throughout the manuscript you state that the dimensions of analysis are sustainability knowledge, behaviour and attitudes. To me the way the dimensions are written is quite confuse.  To me is quite difficult to undersand, for example, what is sustainability attitude. I recommend the authors to think on redifining the terms. I would propose, instead, knowledge on sustainability, and behaviour and attitudes towards sustainability.

Thank you so much for pointing this out. We intended – as you proposed – to investigate students’ development in knowledge regarding sustainability, and behaviour and attitudes towards sustainability. To support readers train of reading, we shortened these descriptions to sustainability knowledge, attitudes, and behaviors. In order to create clarity, we have changed things in two points: 1. In the introduction, we defined the constructs (line 54 ff.).

2. In the theory part we have added two structuring passages, one for knowledge and one for attitudes and behaviour, to create more clarity (line 95 and 128).